nanotechnology

bimodal mesoporous silica, nanoparticle, pH-sensitive, controlled-release, pesticide

**Authors for correspondence:**
Fang Zhang
e-mail: zhangfang2000@bjut.edu.cn
Shiyang Bai
e-mail: sybai@bjut.edu.cn

†These authors contributed equally to this work.

This article has been edited by the Royal Society of Chemistry, including the commissioning, peer review process and editorial aspects up to the point of acceptance.

# pH-sensitive thiamethoxam nanoparticles based on bimodal mesoporous silica for improving insecticidal efficiency

Wenjing Li†, Qi Wang†, Fang Zhang, Hui Shang, Shiyang Bai and Jihong Sun

Faculty of Environment and Life, Beijing University of Technology, Beijing 100124, People's Republic of China

FZ, 0000-0003-1908-3865

In this study, we synthesized pH-sensitive thiamethoxam-3-(2-aminoethylamino) propyl-bimodal mesoporous silica (P/Thi-NN-BMMs) nanoparticles (NPs). We used this bimodal mesoporous silica (BMMs) mesoporous material as a carrier based on the principle of free radical polymerization. The size of the P/Thi-NN-BMMs NPs was about 891.7 ± 4.9 nm, with a zeta potential of about −25.7 ± 2.5 mV. X-ray powder diffraction analysis, $N_2$-sorption measurements and thermogravimetric analysis indicated that thiamethoxam (Thi) was loaded into the pores of the mesoporous structure and that the mesopore surface was coated with polyacrylic acid (PAA). The loading rate of P/Thi-NN-BMMs was about 25.2%. The controlled-release NPs had excellent anti-photolysis performance and storage stability. The NPs showed significant pH sensitivity, and the Thi release rate in pH 10.0 phosphate buffer was higher than those in pH 7.4 and pH 3.0 phosphate buffers. We described the sustained-release curves according to the Weibull model. The relative toxicity of P/Thi-NN-BMMs against peach aphid was 1.44 times that of commercial Thi. This provides a promising instrument for effective insect control and environment protection.

## 1. Introduction

The use of pesticides is the most effective way to protect plants from pests, fungi and weeds in modern agriculture [1]. Low utilization rates and short duration, however, are key issues in the use of pesticides [2–4]. Traditional pesticides can more easily

enter the environment and cannot be easily removed from aqueous solutions [5,6]. Because of environmental and human health concerns, it is urgent to develop new approaches to control loss and improve the efficient utilization of pesticides [4].

Nanotechnology has growing application potential in the field of plant protection [2,6]. Nanomaterials, because of their small size, large specific surface area and ease of modification, enable the efficient use of pesticides. Rapid developments in the field of pesticide research have motivated researchers to develop target-specific nano-pesticides that are less harmful to the environment and that do not reduce efficacy [7,8]. Bimodal mesoporous silica (BMMs), a new kind of mesoporous material with a worm-like pore of about 3 nm in a double-channel structure and a spherical particle-stacking hole of about 10–30 nm, has been used widely as a nanocarrier in the field of biotechnology [9,10]. This material is different from single-cell mesoporous materials and has many unique properties, such as adjustable structure, high chemical and thermal stabilities, environment friendliness and low toxicity. The pore volume of BMMs is significantly higher than that of traditional mesoporous materials, which affords larger pesticide molecules easier accessibility to the active site and significantly improves the loading and release rates of pesticides [9]. Further, surface modification enables the controllable release of specific pesticide molecules with good specificity [11–13]. The stimulus-responsive release system releases the pesticide in response to changes in pH in the environment to effectively increase the pesticide availability [14,15]. Polyacrylic acid (PAA) is a weakly acidic polymer with a carboxyl group. Under alkaline conditions, the ionization state of the polymer changes, resulting in a change in water solubility [16–18]. The digestive systems of insects exhibit alkaline pH [19,20]. Most of the commercially available pesticides are dispersants with particle size of micrometres [21]. Compared with conventional pesticides, nano-pesticide has a greater dispersity due to its smaller particle size and better permeability into the epidermis of pests. Therefore, pH-sensitive release nano-insecticides can offer controlled release in target insects to reduce pesticide loss and improve the efficiency of pesticides [14,22,23].

Thiamethoxam (Thi) is a highly effective and low-toxicity neonicotinoid insecticide that is characterized by contact activity, systemic conductivity and permeability to insects [23]. The use of Thi to control pests is limited, however, because of the short effective period and low effective utilization rates [24,25]. In addition, Thi is highly toxic to bees and other living organisms [26]; therefore, it is important to develop facile approaches to reduce the loss of Thi and enhance utilization efficiency [25]. In this study, we constructed pH-sensitive highly efficient sustained-release system thiamethoxam-3-(2-aminoethylamino) propyl-bimodal mesoporous silica (P/Thi-NN-BMMs) nanoparticles (NPs) for pesticide Thi delivery. The procedure is shown in figure 1. We performed X-ray powder diffraction (XRPD) analysis, $N_2$-sorption measurement and thermogravimetric (TG) analysis and studied various characteristics of P/Thi-NN-BMMs NPs, including the particle size, morphology, XRPD sustained-release properties, storage stability and photolytic properties. We described kinetic analyses according to the Weibull model. We further investigated biological activity of P/Thi-NN-BMMs NPs through an indoor virulence test.

# 2. Material and methods

## 2.1. Chemicals

We purchased acrylic acid (AA) from Tianjin Fuchen Chemical Reagent Factory (Tianjin, China). Azobisisobutyronitrile (AIBN, AR), cetyltrimethylammonium bromide (CTAB, AR), ammonia (25%, AR) and ethyl orthosilicate (TEOS, AR) were purchased from Sinopharm Chemical Reagent Co., Ltd. (Beijing, China). Anhydrous ethanol (AR) and hexane (AR) were purchased from Beijing Chemical Plant (Beijing, China). We also purchased 3-(2-aminoethylamino) propyl-trimethoxysilane (NN-TES, 95%, AR) from Alfa Aesar Ltd (Lancashire, UK). Phosphoric acid ($H_3PO_4$, AR) and disodium hydrogen phosphate ($Na_2HPO_4 \cdot 12H_2O$, AR) were purchased from Tianjin Fuchen Chemical Reagent Factory (Tianjin, China). Technical Thi (97% effective content) and Thi wettable powder (25% effective content) were purchased from Shandong Keda Venture Biotechnology Co., Ltd (Shandong, China). A dialysis bag (molecular weight cut-off 3500) was purchased from Beijing Kebiquan Biotechnology Co., Ltd (Beijing, China).

## 2.2. Preparation of BMMs

First, we weighed 2.612 g of CTAB and dissolved it in 104 ml of distilled water. Afterwards, we slowly added 8 ml of TEOS and then quickly added 3.5 ml of ammonia water with continued stirring until the

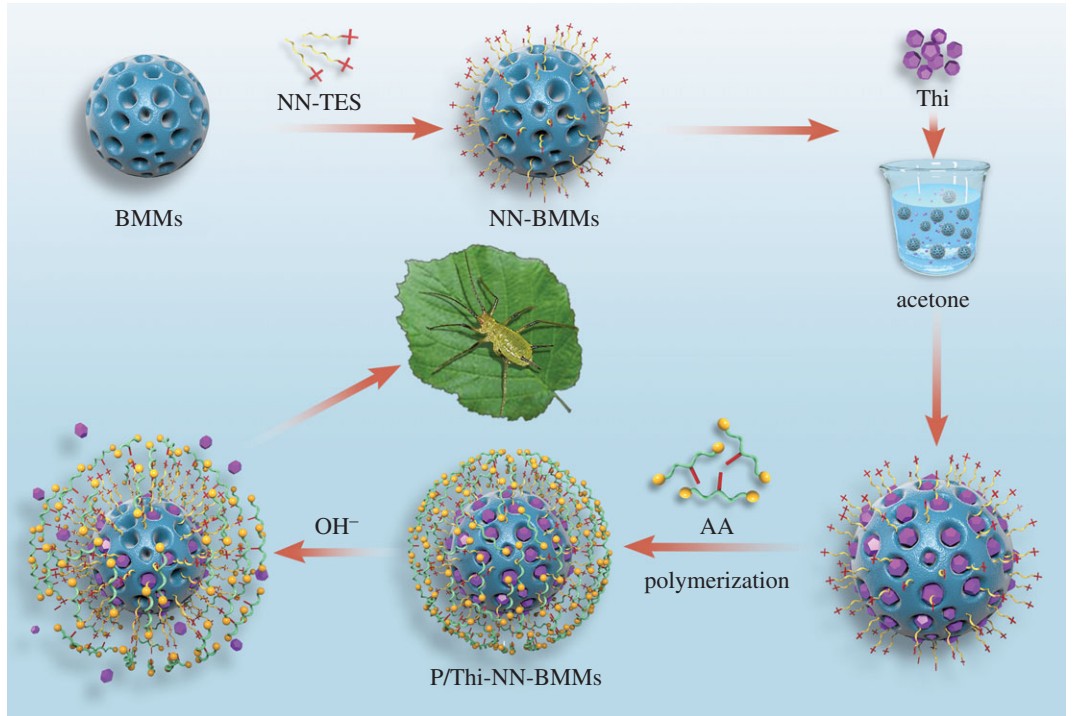

**Figure 1.** Schematic illustration of the construction of P/Thi-NN-BMMs NPs.

solution turned into a white gel. The white gel was suction-filtered and thoroughly washed with distilled water, followed by drying at 120°C for 3 h to obtain a nanomaterial raw powder. The obtained raw powder was heated to 550°C at 5°C min$^{-1}$ for 5 h to remove the organic template to yield BMMs.

## 2.3. Preparation of NN-BMMs and P/NN-BMMs

We placed the BMMs in a vacuum oven. After activation at 120°C for 5 h, 1.0 g of activated BMMs and 0.3 ml of NN-TES were dispersed in 100 ml of toluene and refluxed at 70°C for 4 h. The system was filtered and NN-BMMs was obtained, which were washed and dried. Afterwards, 0.5 g of NN-BMMs, 3 ml of AA and 0.03 g of AIBN were dispersed in 60 ml of ethanol, and then the mixture was refluxed at 70°C for 10 h. We then filtered the system to yield P/NN-BMMs, which were washed and dried at 40°C.

## 2.4. Preparation of Thi-BMMs and P/Thi-NN-BMMs

First, 0.5 g of BMMs was dispersed in 80 ml of Thi acetone solution with a Thi concentration of 30 mg ml$^{-1}$. The system was then stirred at room temperature for 48 h, suction-filtered and washed with anhydrous acetone. Finally, the sample was vacuum-dried, and the nanocomposite Thi-BMMs was obtained. A combination of 0.5 g of NN-BMMs, 2.4 g of Thi and 3 ml of AA was dispersed in 80 ml of acetone, and the mixture was shaken (150 r.p.m.) at room temperature for 48 h to form a homogeneous suspension. The mixture was then refluxed at 70°C for 10 h and washed with acetone. Next, we added 0.03 g of AIBN to the solution to initiate the generation of free radicals by AA. We conducted the whole reaction process under nitrogen protection at 70°C for 20 h. After the system was dried in vacuum at 60°C overnight, we obtained P/Thi-NN-BMMs NPs.

## 2.5. Characterization of P/Thi-NN-BMMs NPs

We observed the morphology of the NPs using a scanning electron microscope (Zeiss Sigma 300, Oberkochen, Germany) with an accelerating voltage of 10 kV. We determined particle size, polymer dispersity index (PDI) and zeta potential of BMMs, NN-BMMs, P/NN-BMMs and P/Thi-NN-BMMs according to dynamic light scattering (DLS) using the Zetasizer Nano ZS (Malvern Instruments Ltd, Malvern, UK). We repeated the test three times and calculated the average value.

## 2.6. X-ray powder diffraction determination

We performed XPRD on an X-ray powder diffractometer (D8 ADVANCE X, Bruker/AXS, Karlsruhe, Germany) using nickel-filtered Cu Kα radiation ($\lambda = 0.154$ nm). The tube voltage and tube current were 35 kV and 35 mA, respectively, and the scan speed was 0.5 min$^{-1}$.

## 2.7. N$_2$-sorption determination

We obtained nitrogen adsorption–desorption isotherms on an Autosorb-iQ pore analyser (Quantachrome, Boynton Beach, FL, USA) at 77 K under continuous adsorption conditions and used Brunauer–Emmett–Teller (BET) and Barrett–Joyner–Halenda analyses to calculate the surface area, pore size and pore volume.

## 2.8. Thermogravimetric analysis

We conducted TG analysis using a thermal analyser (PerkinElmer, Waltham, MA, USA), under the following conditions: nitrogen atmosphere, a flow rate of 20 ml min$^{-1}$ and a heating rate of 10°C min$^{-1}$; the highest temperature was 800°C. The pesticide loading content of the P/Thi-NN-BMMs nano-pesticide was calculated according to the TG curve.

## 2.9. Photolysis behaviour of P/Thi-NN-BMMs NPs

We tested the photolytic properties of P/Thi-NN-BMMs NPs with technical Thi as the control. The samples were divided equally into culture dishes and irradiated under a UV lamp (500 W) at a distance of 20 cm in a UV–light incubator (WFH-203B, Shanghai Precision Instrument Co., Ltd, Shanghai, China). At a given irradiation time (12, 24, 36, 48, 60 and 72 h), we separately collected the culture dishes and analysed the Thi contents in the samples by high-performance liquid chromatography (HPLC). A gradient of methanol : water (40 : 60) was programmed over 30 min at a flow rate of 1 ml min$^{-1}$ with detection at 240 nm. We performed all experiments in triplicate.

## 2.10. Storage stability of P/Thi-NN-BMMs NPs

We determined the stability of the P/Thi-NN-BMMs NPs according to CIPAC MT 46 (Miscellaneous Techniques and Impurities: MT 46 Accelerated Storage Procedure, Collaborative International Pesticides Analytical Council) and GB/T 19136-2003 (Determination of Heat Storage Stability of Pesticides, Chinese National Standard). We stored the samples at $4 \pm 2$°C and $25 \pm 2$°C for 7 days and $54 \pm 2$°C for 14 days and analysed the Thi content using HPLC.

## 2.11. The sustained-release performance of P/Thi-NN-BMMs NPs

First, we suspended 30 mg of P/Thi-NN-BMMs in 100 ml of an ethanol : water mixture (30 : 70, v/v), which was used as the release medium. Then, we collected 2 ml of the sample for testing at fixed time intervals, and the system was replenished with an equal amount of fresh medium. We measured Thi concentrations using HPLC and calculated the release rate of Thi from the nano-delivery sample. The liquid-phase detection conditions included a mobile phase of methanol : water (volume ratio 40 : 60), injection volume of 10 µl, wavelength of 240 nm and retention time of 4 min. The cumulative release rate ($Q$) of Thi in NPs was calculated by the following equation:

$$Q = \frac{V_0 \times C_T + V \times \sum_{N-1}^{T-1} C}{W} \times 100\%,$$
(2.1)

where $W$ is the mass concentration of Thi in NPs, $V_0$ is the volume of the sustained-release solution and $V$ is the volume of the release medium taken at a specific interval time. $C_T$ and $C$ are the mass concentration of $T$ and $N$ samples, respectively. The data of the dynamic Thi sustained release from P/Thi-NN-BMMs NPs were fitted with the pseudo-zero-order equation, pseudo-first-order equation, Higuchi model and Weibull model.

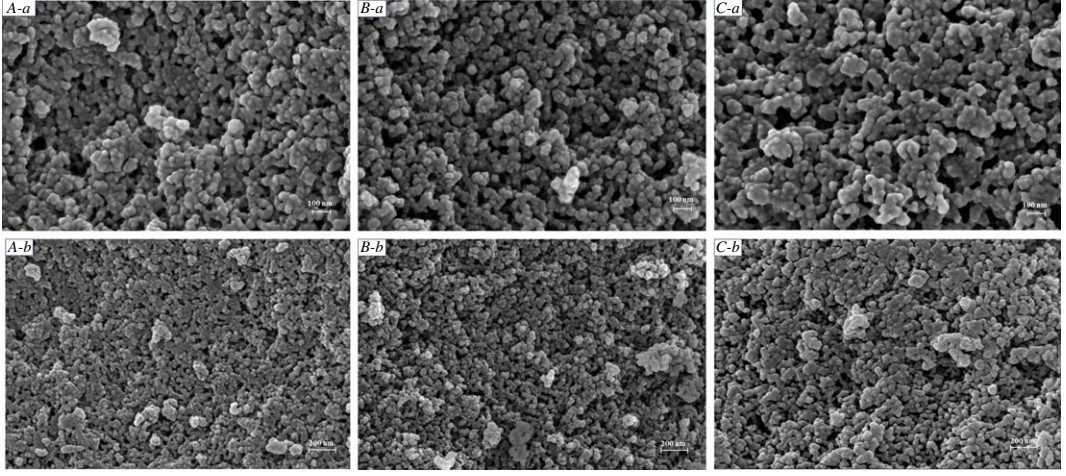

**Figure 2.** SEM images of different NPs. BMMs (*A-a*, *A-b*); NN-BMMs (*B-a*, *B-b*); P/Thi-NN-BMMs (*C-a*, *C-b*).

## 2.12. Evaluation of biological activity of P/Thi-NN-BMMs NPs

We used leaves immersed in chemical for the indoor virulence test of the nano-pesticide loading system. The Thi wettable powder, Thi-BMMs and P/Thi-NN-BMMs were mixed into five series of concentrated liquids with deionized water (400, 200, 100, 50 and 25 µg ml$^{-1}$). The treatment without the agent was used as the control. Fresh cabbage leaves were sliced into 3 cm diameter discs, and 2 ml of 1.5% agar was then added to the bottom of each well of the cell culture plate. Peach aphids were placed into the wells of the prepared leaf, 20 per dish and the well was sealed with rice paper. The treated peach aphids were placed in an incubator with a relative humidity of 75% at 25°C and an illumination condition of 16:8 (L:D). After 24 h, we examined the viability of the insects under a dissecting microscope. Moving insects were considered viable. We calculated the virulence regression equation and the lethal concentration 50 (LC$_{50}$) and its 95% confidence interval using SPSS statistical software. The ratio of the Thi wettable powder to the LC$_{50}$ of P/Thi-NN-BMMs and Thi-BMMs was the relative virulence. We repeated the test three times.

# 3. Results and discussion

## 3.1. Characterization of P/Thi-NN-BMMs NPs

The scanning electron microscopy (SEM) images of different NPs are displayed in figure 2. The nanoscale spherical morphologies of NN-BMMs and P/Thi-NN-BMMs NPs were similar to that of BMMs, indicating that the amino functionalization, the loading of Thi, and the encapsulation with the polymer PAA did not destroy the BMMs morphology. The DLS data (table 1) showed that the hydrated particle size of BMMs was 497.6 ± 7.8 nm, and the size of the amino group-modified particles was further increased to 588.3 ± 4.3 nm. The particle size of P/NN-BMMs, the control, was 768.9 ± 5.1 nm. After Thi was loaded and the mesopore surface was covered with PAA, the size of the hydrated NPs increased to 891.7 ± 4.9 nm (figure 3). The P/Thi-NN-BMMs NPs had the highest PDI value (0.105 ± 0.05), demonstrating a high degree of monodispersity and stability. The positive charge of the amino group in the aqueous solution facilitated electrostatic interaction between the amino group and the negatively charged PAA. The zeta values of the surface-coated PAA-loaded P/NN-BMMs and P/Thi-NN-BMMs were −19.5 ± 1.6 and −25.7 ± 2.5 mV, respectively, which were related to the negatively charged carboxyl groups in the PAA molecule.

We performed XRPD analysis to investigate the crystal structure of nanocapsule. In the XRPD spectrum (figure 4*a*), a characteristic peak occurred at $2\theta = 2.03$ (110), which could be a distinct crystal face diffraction peak of BMMs, indicating that the NP had a highly ordered mesoporous structure [27]. The spectrum of the NN-BMMs modified with the terminal amino group of NN-TES exhibited a similar characteristic peak, suggesting that NN-BMMs still maintained an ordered mesoporous structure. The strength of the XRPD peaks of the PAA-coated NN-BMMs decreased in (110) crystal faces, which showed that the PAA polymer was introduced into the system and that the polymer had

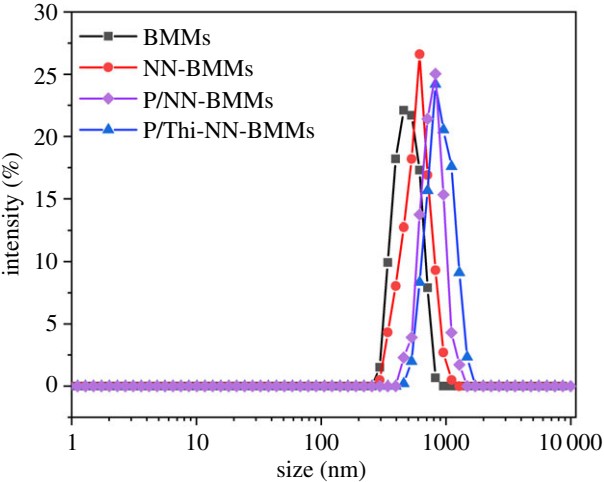

**Figure 3.** Particle size distributions of BMMs, NN-BMMs, P/NN-BMMs and P/Thi-NN-BMMs.

**Table 1.** Mean size, zeta potential and polydispersity index (PDI) of the samples.

| samples | mean size (nm) | zeta potential (mV) | PDI |
| --- | --- | --- | --- |
| BMMs | 497.6 ± 7.8 | 15.3 ± 1.5 | 0.062 ± 0.03 |
| NN-BMMs | 588.3 ± 4.3 | 29.3 ± 2.1 | 0.098 ± 0.02 |
| P/NN-BMMs | 768.9 ± 5.1 | −19.5 ± 1.6 | 0.079 ± 0.03 |
| P/Thi-NN-BMMs | 891.7 ± 4.9 | −25.7 ± 2.5 | 0.105 ± 0.05 |

a significant effect on the order degree of the mesoporous structure. Moreover, regarding the PAA-coated NN-BMMs loaded with Thi, that is, P/Thi-NN-BMMs, the (100) diffraction peak of the spectrum was significantly decreased. The $2\theta$ angle increased (from 1.98° to 2.03°), and the corresponding $d$ value decreased from 44.55 to 43.67 nm, indicating that the encapsulation by the PAA polymer and the occupation of Thi inside the mesopore channels affected the order of the mesoporous structure and reduced the pore size. In summary, this result revealed that Thi was loaded into the NN-BMMs mesoporous structure and that the mesopore surface was coated with PAA.

We determined the loading rate of the P/Thi-NN-BMMs NPs by TG analysis. Figure 4$b$ displays the TG curves of P/Thi-NN-BMMs NPs and the control samples. The weight loss of P/Thi-NN-BMMs was more remarkable than those of NN-BMMs and P/NN-BMMs. The significant weight loss of NPs occurred within the temperature range of 150°C−800°C. The weight loss before 150°C was probably the result of the loss of physically adsorbed water and residual solvent in the channels. The second weight-loss peak occurred at 150–290°C, and the weight loss rate of P/Thi-NN-BMMs NPs was 28%; this weight loss was caused mainly due to the decomposition of the thiazide and terminal amino groups of NN (2.8%) in the mesopores. The carboxyl group of PAA was decarboxylated to form acid anhydride, and a small amount of amino group was partially decomposed at 290–455°C. The last weight-loss peak occurred between 455 and 800°C and was mainly due to the PAA decomposition. As a result, the loading rate of P/Thi-NN-BMMs was about 25.2%.

Furthermore, we used the nitrogen adsorption–desorption technique to characterize the porosity, total pore volume and pore size distribution of NPs. As shown in figure 4$c$, the $N_2$ adsorption–desorption isotherms of BMMs, NN-BMMs, P/NN-BMMs and P/Thi-NN-BMMs conformed to the Langmuir I–V features. The isotherms of BMMs showed two hysteresis loops. The first loop occurred at the relative pressure of $0.3 < P/P_0 < 0.5$, because of the capillary agglomeration of graded pores. The second loop, at $0.85 < P/P_0 < 1.0$, was much steeper and was due to the particle accumulation pores. This finding is consistent with previous reports [9]. As shown in the isotherms, the adsorption process of P/Thi-NN-BMMs was slower than those of BMMs, NN-BMMs and P/NN-BMMs, indicating that the Thi was successfully grafted into the mesoporous channels [28]. The corresponding pore size distribution revealed that all the samples had a dual-model structure and two pore sizes (figure 4$c$, inset). The BET surface area and the pore volume of the BMMs were largest: 995 m$^2$ g$^{-1}$ and

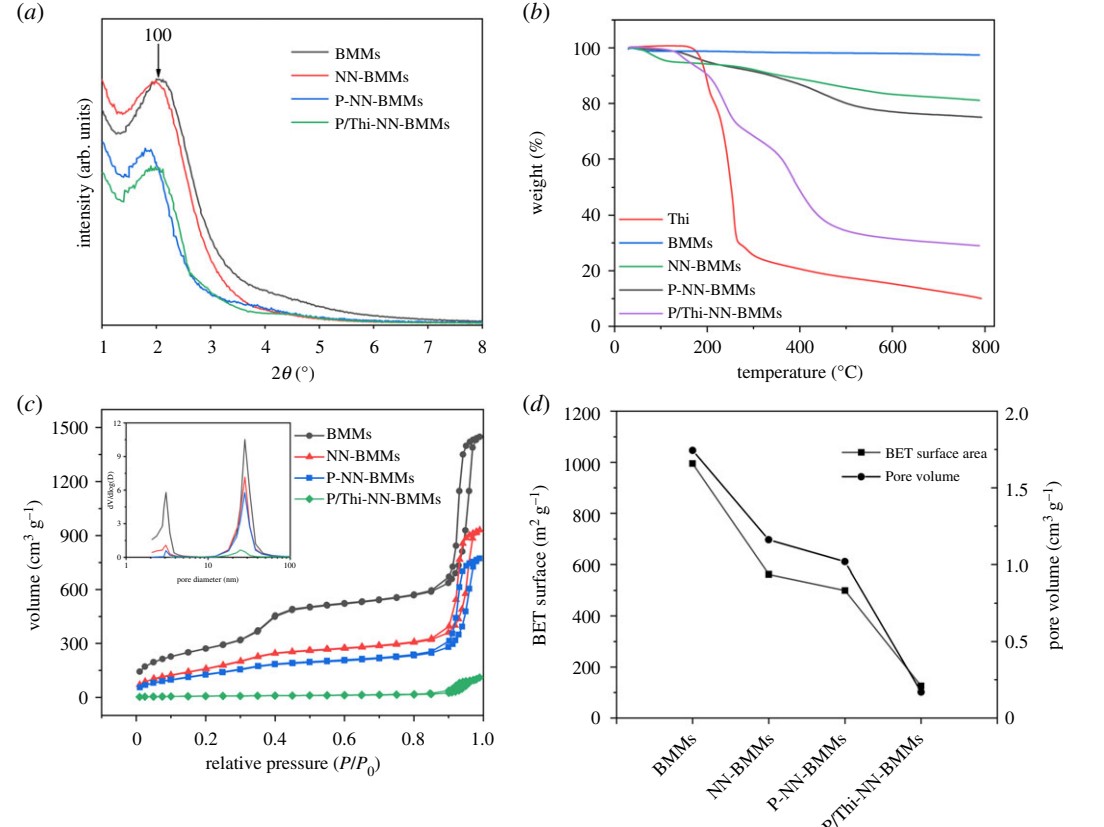

**Figure 4.** XRPD patterns (*a*), TG curves (*b*), $N_2$ adsorption/desorption isotherms (*c*), and BET surface areas and pore volumes (*d*) of P/Thi-NN-BMMs and the control samples.

1.744 cm$^3$ g$^{-1}$, respectively. The specific BET surface area and the pore volume of the P/NN-BMMs NPs (i.e. the BMMs modified with amino groups and encapsulated with the polymer PAA and Thi) were lower, 126 m$^2$ g$^{-1}$ and 0.17 cm$^3$ g$^{-1}$, respectively (figure 4*d*). This result further suggested that the polymers PAA and Thi successfully interacted with the amino groups on the surface of the NN-BMMs channel, resulting in the change of the pore volume.

## 3.2. Photolysis property of P/Thi-NN-BMMs NPs

To evaluate the UV-shielding properties of P/Thi-NN-BMMs NPs, we obtained a photolytic rate curve considering different UV exposure times. The UV exposure was conducted using artificial irradiation. As shown in figure 5, the photolytic percentages of P/Thi-NN-BMMs NPs and technical Thi were 17.89% and 6.86% at 24 h, respectively. The degradation rate of technical Thi was more than 35% after 48 h of exposure to continuous UV light. At 72 h, the photolytic percentages of P/Thi-NN-BMMs NPs and the technical Thi were 19.22% and 42.15%, respectively. These findings showed that the NPs could significantly prevent the photolysis of Thi owing to the protective effect of the BMMs carrier wall.

## 3.3. Storage stability of P/Thi-NN-BMMs NPs

We also evaluated the storage stability of P/Thi-NN-BMMs NPs by measuring the Thi loading contents at different temperatures (figure 6). The result showed that the NP remained stable during storage at room temperature and low temperatures. After 14 days of storage at 54°C, the Thi content was slightly reduced, by 4.35%, which was due to the Thi degradation at a higher temperature. Therefore, the P/Thi-NN-BMMs NPs exhibited good storage stability.

## 3.4. Release characteristics of P/Thi-NN-BMMs NPs

We determined the release effect of P/Thi-NN-BMMs nano-pesticide under different pH conditions. As shown in figure 7, the release rate of P/Thi-NN-BMMs at pH 10.0 was 35.4% after 5 h. Those at pH 7.4

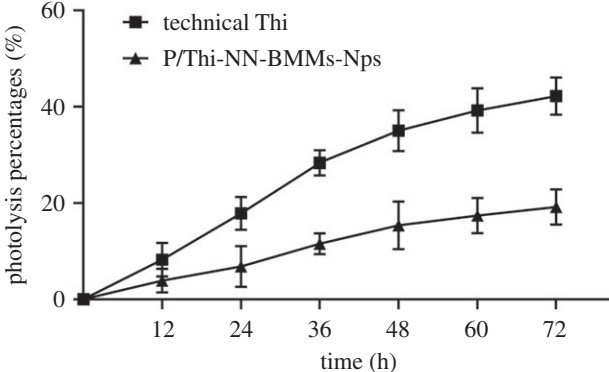

**Figure 5.** Photodegradation curves of P/Thi-NN-BMMs Nps and technical Thi under UV irradiation.

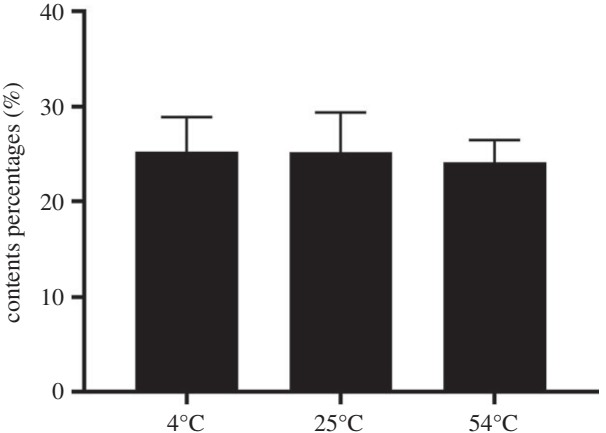

**Figure 6.** Stability of P/Thi-NN-BMMs NPs at different storage temperatures after 14 days.

and pH 3.0 were 26.8% and 23.5%, respectively. The gap was more significant after longer times. After 72 h, the release rate of P/Thi-NN-BMMs nano-pesticide at pH 10.0 was 75.2%, and those at pH 7.4 and pH 3.0 were 54.5% and 49.8%, respectively. The results indicated that the P/Thi-NN-BMMs mesoporous nanocomposites had better pH sensitivity, and the cumulative release rate of the pesticide under alkaline conditions was higher than that under acidic and neutral conditions. This was because PAA covered the surface of the mesoporous structure, and the acidity coefficient p$Ka$ of –COOH in PAA was 4.25. In the alkaline environment, the carboxyl group in the PAA was easily ionized, so that the PAA wrapped on the surface of the mesoporous composite material exhibited a 'loose' state; thus, the Thi in the sample was released relatively easily. Conversely, the carboxyl groups in the PAA were not ionized under acidic conditions, and the hydrogen bond between the –COOH forced the PAA on the surface of the composite to 'shrink,' thereby limiting the Thi release.

## 3.5. Study of release kinetics

To further elucidate the effect of pH on the mechanism of Thi sustained release from P/Thi-NN-BMMs NPs, we studied the release kinetics using the Higuchi model [29] and Weibull kinetic model [30]. Table 2 presents the values of parameters and the regression coefficient ($R^2$). The data of the sustained release of Thi were fitted to zero-order and first-order equations, the Higuchi kinetic model and the Weibull model. The data of the Weibull model better described the sustained release than the Higuchi model, as indicated by the higher $R^2$, which showed that the release behaviour of P/Thi-NN-BMMS NPs was consistent with the Weibull equation. In addition, the $a_0$ values at pH 3.0, 7.4 and 10.0 were 0.918, 0.788 and 0.829, respectively, and were distributed in the range of 0.75–1, which corresponded with the combined mechanism of Fickian diffusion and transport Case II [31].

## 3.6. Evaluation of the insecticidal effect of P/Thi-NN-BMMs NPs

To evaluate the insecticidal activity of P/Thi-NN-BMMs NPs, we investigated the effects of NPs on the peach aphid. The LC$_{50}$ values of P/Thi-NN-BMMs and Thi-BMMs were 83.949 and

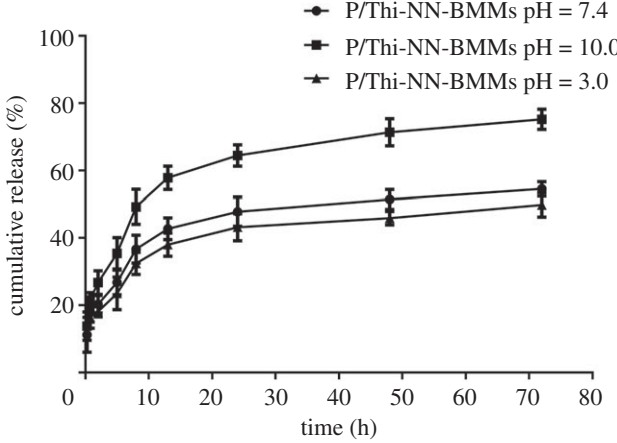

**Figure 7.** Cumulative release profiles of P/Thi-NN-BMMs with pH of 3.0, 7.4 and 10.0.

**Table 2.** Kinetic parameters of the Thi released from P/Thi-NN-BMMs with pH of 3.0, 7.4 and 10.0. $y$ is the fractional release of pesticide; $x$ is the elapsed time; $a_0$ is the kinetic release constant and $a_1$ is constant. $R^2$ is the high value of linear regression coefficient.

| fitting model | formula | pH | $a_0$ | $a_1$ | $R^2$ |
|---|---|---|---|---|---|
| zero-order | $y = a_0 x$ | 3.0 | 0.758 | / | 0.701 |
|  |  | 7.4 | 0.525 | / | 0.688 |
|  |  | 10.0 | 0.477 | / | 0.717 |
| first-order | $y = a_0(1 - \exp(-a_1 x))$ | 3.0 | 0.179 | 39.046 | 0.803 |
|  |  | 7.4 | 0.189 | 50.145 | 0.836 |
|  |  | 10.0 | 0.168 | 69.539 | 0.879 |
| Higuchi | $y = a_0 x^{0.5}$ | 3.0 | 4.839 | / | 0.898 |
|  |  | 7.4 | 5.383 | / | 0.880 |
|  |  | 10.0 | 7.744 | / | 0.891 |
| Weibull | $y = 1 - \exp(-(x_0^a)/a_1)$ | 3.0 | 0.918 | 10.881 | 0.983 |
|  |  | 7.4 | 0.788 | 10.207 | 0.988 |
|  |  | 10.0 | 0.829 | 10.687 | 0.992 |

**Table 3.** Insecticidal activity of Thi against peach aphid after 24 h.

| pesticide type | virulence regression equation | $LC_{50}$ ($\mu g\ ml^{-1}$) | 95% confidence interval | relative virulence |
|---|---|---|---|---|
| Thi wettable powder | $y = -2.606 + 1.251x$ | 121.267 | 72.981–224.801 | 1 |
| P/Thi-NN-BMMs | $y = -3.936 + 2.046x$ | 83.949 | 59.050–116.405 | 1.44 |
| Thi-BMMs | $y = -3.944 + 1.989x$ | 96.118 | 67.866–135.250 | 1.26 |

96.118 µg ml$^{-1}$, respectively, which were 30.77% and 20.73% lower than that of Thi wettable powder (table 3). The relative virulence values of P/Thi-NN-BMMs and Thi-BMMs were 1.44 and 1.26 times that of Thi wettable powder, respectively. Nano-pesticide formulations have a smaller particle size and larger specific surface area, which can enhance their contact with tissues, such as insect epidermis. Therefore, they may be more effective against target insects than traditional pesticide formulations. The LC$_{50}$ of P/Thi-NN-BMMs nano-pesticide was lower than that of Thi-BMMs, which may be related to the better release effect of pests under alkaline conditions. The peach aphid has an

alkaline digestive system, and P/Thi-NN-BMMs nano-pesticide can be released gradually in the peach aphid because of the pH-responsive and sustained-release properties of NPs. In conclusion, the results showed that P/Thi-NN-BMMs had stronger insecticidal activity than Thi-BMMs and commercially available formulations.

# 4. Conclusion

In this study, we successfully constructed pH-sensitive sustained-release P/Thi-NN-BMMs NPs based on a bimodal nano-mesoporous material BMMs loaded with Thi and modified pH-sensitive PAA polymer. The P/Thi-NN-BMMs NPs showed good water dispersibility and monodispersity. The characterization analyses confirmed that the NPs modified by functionalized groups displayed the structural integrity of the BMMs mesopore structure and had a remarkable loading rate of 25.2%. Moreover, P/Thi-NN-BMMs NPs exhibited good anti-photolysis property and storage stability. The sustained-release experiments showed that the NPs had excellent pH stimuli-responsive properties, and the cumulative release of Thi in an alkaline environment (pH 10.0) was higher than those in neutral (pH 7.4) and acidic (pH 3.0) environments. The Weibull kinetic model could describe the sustained-release curves, and the model was consistent with the Fickian diffusion mechanism. The Thi nano-delivery system displayed a higher insecticidal efficacy against peach aphid compared with the technical Thi. This study provided a new approach to improve pesticide efficacy, reduce the residue of organic solvents and promote environmental protection and human health.

Data accessibility. Data available from the Dryad Digital Repository: https://doi.org/10.5061/dryad.79cnp5ht5.

Authors' contributions. F.Z. and S.B. designed the study and interpreted the results; Q.W., W.L. and H.S. conducted the researches; F.Z., S.B. and J.S. analysed data; F.Z., Q.W. and W.L. wrote the paper.

Competing interests. We declare we have no competing interests.

Funding. This work was supported by National Key R&D Program of China (grant no. 2016YFD0200502-2).

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
