## [Peer Review File · Royal Society Open Science]

Review History

RSOS-200604.R0 (Original submission)

Review form: Reviewer 1

Is the manuscript scientifically sound in its present form?

Yes

Are the interpretations and conclusions justified by the results?

Yes

Is the language acceptable?

Yes

Do you have any ethical concerns with this paper?

Yes

Have you any concerns about statistical analyses in this paper?

No

Recommendation?

Major revision is needed (please make suggestions in comments)

Comments to the Author(s)

General comments:

In this work, pH-sensitive thiamethoxam-3-(2-aminoethylamino) propyl-double model mesoporous silica (P/Thi-NN-BMMs) nanoparticles was served as the carrier based on the principle of free radical polymerization. The topic and results are interesting. However, this manuscript should have major revision before published. My comments are:

1. Introduction: Polymer-coated nanomaterials and mesoporous silica applied in sustained release carrier has been widely reported, and several recent reports are recommended to be included in the Introduction part.

For examples:

1. Royal Society Open Science, 2018, 5(7): 0-180658
2. Journal of Polymers & the Environment, 2018:1-9.
3. Journal of Agricultural and Food Chemistry, 2016, 64, 8095-8102.

2. If possible, the sustained release curve should be fitted with different kinetics model to illustrate the mechanism of their sustained release performance.

3. Why do you choose pH 7.4 instead of neutral environment when studying the effect of pH value on the sustained release performance of the samples..

4. The BET surface and pore data results of Nitrogen adsorption-desorption isotherms should be listed.

5. The English of the manuscript should be improved and the mistakes of expression should be revised before publication. For example, "indicating that the influence, indicating that the amino functionalization, the loading of thiamethoxam." "This indicated that encapsulation of the PAA polymer and Thi loading affected the order of the mesoporous structure and reduced the pore size." "These materials had better control against peach aphids."

6. In the experiment of RXRD, what did "d value" refer to?

7. What did LC50 refer to? The amount of thiamethoxam actually contained in the carrier or the total amount of the sample.

8. How to calculate the pesticide loading content of the P/Thi-NN-BMMs nano-pesticide while NN-BMMs still had weigh loss in TG curves? Why not use HPLC as performed in the test for the storage stability of P/Thi-NN-BMMs Nps.

Review form: Reviewer 2

Is the manuscript scientifically sound in its present form?

No

Are the interpretations and conclusions justified by the results?

No

Is the language acceptable?

No

Do you have any ethical concerns with this paper?

No

Have you any concerns about statistical analyses in this paper?

No

Recommendation?

Reject

Comments to the Author(s)

The article is quite short. Introduction is really short. It will be nice if authors explain: i) why they have selected this nanoparticles, ii) why they coated them, iii) why they use Thi as active ingredient, and iv) what is the problem and/or target. Further there are some important missing references: for example, during the introduction, experimental methods. Authors need to address this issue.

During the abstract, there are some abbreviations that need to be explained during the abstract (i.e. BMMs). They need to add particle size and z-potential average. In fact, why they are performing DLS in water? It is like a simulated insect gastrointestinal media?

During the introduction English must be improve. It is quite difficult to follow the discussion. There are some abbreviations that need to be explained during the introduction (i.e. THi) and it has to be extended.

No references about what a BBM is. Add some references as it is impossible to consider if they are good candidates for the release of pesticides or not, and to further compare with other porous silica materials. Add references here:

“Double model mesoporous silica materials (BMMs) are a new type of mesoporous material with a worm-like pore of about 3 nm in a double-channel structure and a spherical particle-stacking hole of about 10-30 nm (REF). BMMs are different from single-cell mesoporous materials and have many unique properties, such as controllable structure and controllable particle size (REF).”

What are the main differences between “Double model mesoporous silica materials (BMMs)” and mesoporous silica? As I know mesoporous can be synthesized with a controllable structure and particle size.

I suppose that the release of Thi is performed at different pH as insect have to eat these particles and then the Thi will be released. But, what is a maximum particle size than a small insect can eat? Add this information in the introduction. Further add references here: “The pH of the digestive tract of herbivorous insects is mostly alkaline.”

In Figure 2, particles are really aggregated and it is quite difficult to distinguish the spherical morphology. Nanoparticles need to be dispersed and better SEM images (or TEM) need to be performed.

DLS results show that particle size increases from 497 to 891 nm once Thi is loaded and the surface is covered with PAA. Is this increment due to the PAA or Thi in the surface? Comment on that. Is the PAA size related with this difference on size ($891 - 497 = 394$ nm). Do you have the characterization of the BBM+Thi or BBM+PAA combinations?

Important characterization regarding the PAA-BMM coated nanoparticles (without Thi is needed) in order to assess the discussion. Authors need to add TGA, DXRP, DLS of PAA-BMM (without Thi)

In the TGA indicate where are the 3 weight losses of Thi-NN-BMMS. Add the TGA of Thi, as it will be easier to compare.

Add DRXP patterns. There is no figure related with this results.

How can the authors calculate the amount of encapsulated Thi? I cannot find this information along the text.

Decision letter (RSOS-200604.R0)

Dear Dr Zhang:

Manuscript ID: RSOS-200604

Title: "pH-sensitive Thiamethoxam Nanoparticles Based on Double Model Mesoporous Silica for Improving Insecticidal Efficiency"

Thank you for submitting the above manuscript to Royal Society Open Science. Your paper was sent to reviewers and their comments are included at the bottom of this letter.

In view of the concerns raised by the reviewers, the manuscript has been rejected in its current form. However, a new manuscript may be submitted which takes into consideration these comments.

Please note that resubmitting your manuscript does not guarantee eventual acceptance, and that your resubmission will be subject to peer review before a decision is made.

Your resubmitted manuscript should be submitted by 27-Oct-2020. If you are unable to submit by this date please contact the Editorial Office.

On behalf of the Subject Editor Professor Anthony Stace and the Associate Editor Dr Dattatray
 Late

REVIEWER(S) REPORTS:

Associate Editor Comments to Author ():

RSC Associate Editor:

Comments to the Author:

(There are no comments.)

RSC Subject Editor:

Comments to the Author:

detail characterization data need to be added

Reviewers' Comments to Author:

Reviewer: 1

Comments to the Author(s)

General comments:

In this work, pH-sensitive thiamethoxam-3-(2-aminoethylamino) propyl-double model mesoporous silica (P/Thi-NN-BMMs) nanoparticles was served as the carrier based on the principle of free radical polymerization. The topic and results are interesting. However, this manuscript should have major revision before published. My comments are:

1. Introduction: Polymer-coated nanomaterials and mesoporous silica applied in sustained release carrier has been widely reported, and several recent reports are recommended to be included in the Introduction part.

For examples:

1. Royal Society Open Science, 2018, 5(7): 0-180658
2. Journal of Polymers & the Environment, 2018:1-9.
3. Journal of Agricultural and Food Chemistry, 2016,64, 8095-8102.

2. If possible, the sustained release curve should be fitted with different kinetics model to illustrate the mechanism of their sustained release performance.

3. Why do you choose pH 7.4 instead of neutral environment when studying the effect of pH value on the sustained release performance of the samples..

4. The BET surface and pore data results of Nitrogen adsorption-desorption isotherms should be listed.

5. The English of the manuscript should be improved and the mistakes of expression should be revised before publication. For example, "indicating that the influence, indicating that the amino functionalization, the loading of thiamethoxam." "This indicated that encapsulation of the PAA

polymer and Thi loading affected the order of the mesoporous structure and reduced the pore size.”” These materials had better control against peach aphids.”

6. In the experiment of RXRD, what did “d value” refer to?

7. What did LC50 refer to? The amount of thiamethoxam actually contained in the carrier or the total amount of the sample.

8. How to calculate the pesticide loading content of the P/Thi-NN-BMMs nano-pesticide while NN-BMMs still had weigh loss in TG curves? Why not use HPLC as performed in the test for the storage stability of P/Thi-NN-BMMs Nps.

Reviewer: 2

Comments to the Author(s)

The article is quite short. Introduction is really short. It will be nice if authors explain: i) why they have selected this nanoparticles, ii) why they coated them, iii) why they use Thi as active ingredient, and iv) what is the problem and/or target. Further there are some important missing references: for example, during the introduction, experimental methods. Authors need to address this issue.

During the abstract, there are some abbreviations that need to be explained during the abstract (i.e. BMMs). They need to add particle size and z-potential average. In fact, why they are performing DLS in water? It is like a simulated insect gastrointestinal media?

During the introduction English must be improve. It is quite difficult to follow the discussion. There are some abbreviations that need to be explained during the introduction (i.e. THi) and it has to be extended.

No references about what a BBM is. Add some references as it is impossible to consider if they are good candidates for the release of pesticides or not, and to further compare with other porous silica materials. Add references here:

“Double model mesoporous silica materials (BMMs) are a new type of mesoporous material with a worm-like pore of about 3 nm in a double-channel structure and a spherical particle-stacking hole of about 10-30 nm (REF). BMMs are different from single-cell mesoporous materials and have many unique properties, such as controllable structure and controllable particle size (REF).” What are the main differences between “Double model mesoporous silica materials (BMMs)” and mesoporous silica? As I know mesoporous can be synthesized with a controllable structure and particle size.

I suppose that the release of Thi is performed at different pH as insect have to eat these particles and then the Thi will be released. But, what is a maximum particle size than a small insect can eat? Add this information in the introduction. Further add references here: “The pH of the digestive tract of herbivorous insects is mostly alkaline.”

In Figure 2, particles are really aggregated and it is quite difficult to distinguish the spherical morphology. Nanoparticles need to be dispersed and better SEM images (or TEM) need to be performed.

DLS results show that particle size increases from 497 to 891 nm once Thi is loaded and the surface is covered with PAA. Is this increment due to the PAA or Thi in the surface? Comment on that. Is the PAA size related with this difference on size ($891 - 497 = 394$ nm). Do you have the characterization of the BBM+Thi or BBM+PAA combinations?

Important characterization regarding the PAA-BMM coated nanoparticles (without Thi is needed) in order to assess the discussion. Authores need to add TGA, DXRP, DLS of PAA-BBM (without Thi)

In the TGA indicate where are the 3 weight losses of Thi-NN-BMMS. Adde the TGA of Thi, as it will be easier to compare.

Add DRXP patterns. There is no figure related with this results.

How can the authors calculate the amount of encapsulated Thi? I cannot find this information along the text.

Author's Response to Decision Letter for (RSOS-200604.R0)

See Appendix A.

RSOS-201967.R0

Review form: Reviewer 1

Is the manuscript scientifically sound in its present form?

Yes

Are the interpretations and conclusions justified by the results?

Yes

Is the language acceptable?

Yes

Do you have any ethical concerns with this paper?

No

Have you any concerns about statistical analyses in this paper?

No

Recommendation?

Accept as is

Comments to the Author(s)

The manuscript has met the standard for publication.

Review form: Reviewer 2

Is the manuscript scientifically sound in its present form?

Yes

Are the interpretations and conclusions justified by the results?

Yes

Is the language acceptable?

Yes

Do you have any ethical concerns with this paper?

No

Have you any concerns about statistical analyses in this paper?

No

Recommendation?

Accept as is

Comments to the Author(s)

Authors addressed all the mentioned issues.

Decision letter (RSOS-201967.R0)

This year has been very difficult for everyone, and we want to take the opportunity to thank you for your continued support in 2020.

The Royal Society Open Science editorial office will be closed from the evening of Friday 18 December 2020 until Monday 4 January 2021. We will not be responding during this time. If you have received a deadline within this time period, please contact us as soon as possible to allow us to extend the deadline. If you receive any automated messages during this time asking you to meet a deadline, we offer apologies and invite you to respond after the festive period or during normal working hours.

With our best for a peaceful festive period and New Year, and we look forward to working with you in 2021.

Dear Dr Zhang:

Title: pH-Sensitive Thiamethoxam Nanoparticles Based on Bimodal Mesoporous Silica for Improving Insecticidal Efficiency
Manuscript ID: RSOS-201967

It is a pleasure to accept your manuscript in its current form for publication in Royal Society Open Science. The chemistry content of Royal Society Open Science is published in collaboration with the Royal Society of Chemistry.

Yours sincerely,
Dr Laura Smith

Publishing Editor, Journals

On behalf of the Subject Editor Professor Anthony Stace and the Associate Editor Dr Dattatray Late.

RSC Associate Editor

Comments to the Author:

Both reviewer are satisfied the changes made in re-submission. I recommend manuscript to be accepted for publication.

Reviewer(s)' Comments to Author:

Reviewer: 2

Comments to the Author(s)

Authors addressed all the mentioned issues.

Reviewer: 1

Comments to the Author(s)

The manuscript has met the standard for publication.

Appendix A

To: Dr. Laura Smith
Editor
Royal Society Open Science

From: Dr. Fang Zhang
Beijing University of Technology
100# Pingleyuan, Chaoyang District, Beijing 100124, P.R. China
E-mail: zhangfang2000@bjut.edu.cn

Subject: Revision Requested: Manuscript ID RSOS-200604

Title: pH-Sensitive Thiamethoxam Nanoparticles Based on Bimodal Mesoporous Silica for Improving Insecticidal Efficiency

Dear Editor:

Thank you very much for your kind letter on April 29th regarding our manuscript entitled “**pH-Sensitive Thiamethoxam Nanoparticles Based on Bimodal Mesoporous Silica for Improving Insecticidal Efficiency**” (Manuscript ID: **RSOS-200604**). We appreciate the prompt review of our paper and your encouraging decision. We appreciate the reviewers’ comments and those comments are all valuable and very helpful for revising and improving our paper. We have revised the paper in accordance with the reviewers’ suggestions. These changes will not influence the content and framework of the paper. We are looking forward to hearing from you regarding our resubmission. We would be glad to respond to any further questions and comments that you may have. Once again, thank you very much for your comments and suggestions.

The main corrections in the paper and the point-by-point responses to the reviewer’s comments are as following.

Yours sincerely,

Dr. Fang Zhang
Beijing University of Technology
E-mail: zhangfang2000@bjut.edu.cn

REVIEWERS' COMMENTS AND AUTHOR' RESPONSES

Note: Our responses (standard typeface) to reviewers' comments (bold); the yellow highlighted words and sentences have been added to the main text.

Reviewer #1:

1. Introduction: Polymer-coated nanomaterials and mesoporous silica applied in sustained release carrier has been widely reported, and several recent reports are recommended to be included in the Introduction part.

For examples:

1.1 Royal Society Open Science, 2018, 5(7): 0-180658

1.2 Journal of Polymers & the Environment, 2018:1-9.

1.3 Journal of Agricultural and Food Chemistry, 2016, 64, 8095-8102.

Response: We really appreciate the thoughtful suggestions made by the reviewer. We supplied some reports about polymer-coated nanomaterials and mesoporous silica applied in sustained release carrier in the introduction part, and added references [9-13] in the revised manuscript.

2. If possible, the sustained release curve should be fitted with different kinetics model to illustrate the mechanism of their sustained release performance.

Response: Thanks for the reviewer's comment. We added the release curve with different kinetics model to illustrate the mechanism of their sustained release performance, and the data was also provided in the revised manuscript (Table 2).

Table 2 Fitting results for sustained release of Thi at different pH values.

Fitting model	formula	pH	a_0	a_1	R^2
Zero-order	$y=aox$	3.0	0.758	/	0.701
		7.4	0.525	/	0.688
		10.0	0.477	/	0.717
First-order	$y=a_1(1-exp(-aox))$	3.0	0.179	39.046	0.803
		7.4	0.189	50.145	0.836
		10.0	0.168	69.539	0.879
Higuchi	$y=aox^{0.5}$	3.0	4.839	/	0.898
		7.4	5.383	/	0.880
		10.0	7.744	/	0.891
Weibull	$y=1-exp((-x^{a_0})/a_1)$	3.0	0.918	10.881	0.983
		7.4	0.788	10.207	0.988
		10.0	0.829	10.687	0.992

Where y was the fractional release of pesticide; x was the elapsed time; a_0 was the kinetic release constant and a_1 was constant. R^2 was the high value of linear regression coefficient.

3. Why do you choose pH 7.4 instead of neutral environment when studying the effect of pH value on the sustained release performance of the samples.

Response: The range of neutral phosphate buffer is usually between 5.5 and 8.5. It has been reported in some literature that phosphate buffer (pH 7.4) was often used as the neutral release medium when studying the effect of pH value on the sustained release performance of nano-pesticides^[1-3]. We chose pH 7.4 instead of neutral environment according to the references:

References:

- [1] Ma HX, Niu YS, Wang M et.al. Synthesis and characterization of pH-sensitive block polymer poly (propylene carbonate)-b-poly (acrylic acid) for sustained chlorpyrifos release. *Mater. Today Commun*, 2020, 24: 2352-4928.
- [2] Li, GB. Wang, J. Kong, XP. Coprecipitation-based synchronous pesticide encapsulation with chitosan for controlled spinosad release. *Carbohydrate Polymers*. 2020, 249:0144-8617
- [3] Xu CL, Cao LD, Zhao PY, et al. Emulsion-based synchronous pesticide encapsulation and surface modification of mesoporous silica nanoparticles with carboxymethyl chitosan for controlled azoxystrobin release. 2018, *Chem Eng J*, 2018, 348: 244–254.

4. The BET surface and pore data results of Nitrogen adsorption-desorption isotherms should be listed.

Response: We greatly appreciate the thoughtful suggestions. According to the reviewer's suggestion, we conducted Nitrogen adsorption-desorption experiments, and the BET surface and pore volume of Nitrogen adsorption-desorption isotherms were provided in the revised manuscript. (Fig 4C, 4D)

5. The English of the manuscript should be improved and the mistakes of expression should be revised before publication. For example, “indicating that the influence, indicating that the amino functionalization, the loading of thiamethoxam.” “This indicated that encapsulation of the PAA polymer and Thi loading affected the order of the mesoporous structure and reduced the pore size.”” These materials had better control against peach aphids.”

Response: We are very grateful for the reviewers' comments. We have carefully revised the

manuscript according to your kind suggestions, and the manuscript was polished by native English speakers.

6. In the experiment of RXRD, what did “d value” refer to?

Response: The “d value” referred to the interplanar spacing, and the “d value” was obtained by analyzing the spectrum. The X-ray diffraction phenomenon was based on the Bragg formula:

$$2d \sin \theta = n\lambda$$

where “d” was the interplanar spacing in a certain direction of the crystal lattice, and θ was the glancing angle in the diffraction experiment. n was an integer and called the interference series.

7. What did LC50 refer to? The amount of thiamethoxam actually contained in the carrier or the total amount of the sample.

Response: Lethal Concentration 50(LC50) referred to the concentration of pesticide that caused 50% of the death of living organism. We added the information in the revised manuscript.

8. How to calculate the pesticide loading content of the P/Thi-NN-BMMs nano-pesticide while NN-BMMs still had weigh loss in TG curves? Why not use HPLC as performed in the test for the storage stability of P/Thi-NN-BMMs Nps.

Response: We greatly appreciate the thoughtful suggestions. Based on previous reports, we calculated the loading rate of Thi by TG analysis^[1-3]. Figure 4B showed the data of TG analysis, which provided qualitative and quantitative information about the physical state of P/Thi-NN-BMMs nanoparticles and the control samples. The weight loss of P/Thi-NN-BMMs was more remarkable than those of NN-BMMs and P/NN-BMMs. The significant weight loss of NPs occurred within the temperature range of 150°C–800°C. The weight loss before 150°C was probably the result of the loss of physically adsorbed water and residual solvent in the channels. The second weight-loss peak occurred at 150°C–290°C, and the weight loss rate of P/Thi - NN-BMMs NPs was 28% (96.9-68.9%=28%). This weight loss was caused mainly due to the decomposition of the Thi and terminal amino groups of NN (2.8%) in the mesopores. As a result, the loading rate of P/Thi-NN-BMMs was about 25.2% (28%-2.8%=25.2%).

We detected the content of Thi in the test for the storage stability of P/Thi-NN-BMMs Nps by HPLC, and added the data in the revised manuscript.

References

- [1] Zhang, WB. He S. Liu Y. 2014 Preparation and Characterization of Novel Functionalized Prochloraz Microcapsules Using Silica-Alginate-Elements as Controlled Release Carrier Materials. *ACS Appl. Mater. Interfaces* 14: 11783-11790. (doi:10.1021/am502541g).
- [2] Gao YH. Zhang YH. He S. et al. 2019 Fabrication of a hollow mesoporous silica hybrid to improve the targeting of a pesticide. *Chem. Eng. J.* 364: 361-369 (doi:10.1016/j.cej.2019.01.105)
- [3] Mainardes R M, Gremião M P D, Evangelista R C. Thermoanalytical study of praziquantel-loaded PLGA nanoparticles. *Rev. Bras. Cienc. Farm.*, 2006,42(2):1516-9332

Reviewer #2:

1. The article is quite short. Introduction is really short. It will be nice if authors explain: i) why they have selected this nanoparticles, ii) why they coated them, iii) why they use Thi as active ingredient, and iv) what is the problem and/or target. Further there are some important missing references: for example, during the introduction, experimental methods. Authors need to address this issue.

Response: We really appreciate the thoughtful suggestions made by the reviewer. We supplemented some related reports in the introduction, and answered these questions in the revised manuscript.

i) why they have selected this nanoparticles?

Response: Bimodal mesoporous silica (BMMs), a new kind of mesoporous material with a worm-like pore of about 3 nm in a double-channel structure and a spherical particle-stacking hole of about 10-30 nm, was widely used as nanocarrier in the field of biotechnology. BMMs was different from single-cell mesoporous materials and had many unique properties, such as adjustable structure, high chemical and thermal stability, environmentally friendly property, and low toxicity. Compared with the traditional mesoporous material, the pore volume of BMMs obvious increased from $0.25 \text{ cm}^3 \text{ g}^{-1}$ to $1.8 \text{ cm}^3 \text{ g}^{-1}$, which allowed the larger pesticide molecules easier accessibility to the active site, and significantly improved the loading/release rates of pesticides.

ii) why they coated them?

Response: Based on the unique dual-channel structure and larger pore volume of BMMs, it maintained good sustained and controlled release performance on the basis of achieving high drug loading^[1,2]. Further surface modification allowed one to perform controllable release of specific pesticide molecules with good specificity. We chose BMMs as nanocarrier, and constructed the pH-sensitive sustained release P/Thi-NN-BMMs nanoparticle.

iii) why they use Thi as active ingredient?

Response: Thiamethoxam(Thi) was a highly effective and low-toxicity class of neonicotinoid insecticides, which had contact activity, systemic conductivity, and permeability to insects. However, the use of Thi in the controlling pests was limited due to the short effective period and

low effective utilization rates. In addition, Thi was highly toxic to bees and other living organisms. Therefore, it was urgent to develop facile approaches to reduce the loss of Thi and enhance the utilization efficiency^[3,4].

iv) what is the problem and/or target?

Response: The use of Thi in the controlling pests was limited due to the short effective period and low effective utilization rates. In addition, Thi was highly toxic to bees and other living organisms.

Reference:

[1]Gao, L.Sun, J. H.Li, Y. Z. Bimodal Mesoporous Silicas Functionalized with Different Level and Species of the Amino Groups for Adsorption and Controlled Release of Aspirin. *J. Nanosci. Nanotechnol.* 2011, 11 6690-6697
 [2].Xiaoqi Jin, Qian Wang, Jihong Sun, Hamida Panezail, Shiyang Bai, Xia Wu, P(NIPAM-co-AA) @BMMs with mesoporous silica core and controlled copolymer shell and its fractal characteristics for dual pH- and temperature-responsive performance of ibuprofen release, *Int. J. Polym. Mater. Polym. Biomater.*2017, 67 : 131-142
 [3] Elabasy A, Shoaib A, Waqas M, et al. Cellulose Nanocrystals Loaded with Thiamethoxam:Fabrication, Characterization, and Evaluation of Insecticidal Activity against Phenacoccus solenopsis Tinsley (Hemiptera: Pseudococcidae). *Nanomaterials* 2020, 10, 788-801.
 [4] Gao YH, Xiao YN, Mao KK,et al. Thermoresponsive polymer-encapsulated hollow mesoporous silica nanoparticles and their application in insecticide delivery.

2. During the abstract, there are some abbreviations that need to be explained during the abstract (i.e. BMMs). They need to add particle size and z-potential average. In fact, why they are performing DLS in water? It is like a simulated insect gastrointestinal media?

Response: Thank you very much. We added some abbreviations in the abstract of the revised manuscript. We determined the particle size and z-potential average, and supplied the data in the **table 1** in the revised manuscript. The nanoparticles had good water solubility, and DLS was detected usually in water according to the previous reports ^[1,2].

Table.1 Mean size , zeta potential and polydispersity index (PDI) of nano-drug delivery system.

Nps	Mean size (nm)	Zeta potential (mV)	PDI
BMMs	497.6±7.8	15.3±1.5	0.062±0.03
NN-BMMs	588.3±4.3	29.3±2.1	0.098±0.02
P/NN-BMMs	768.9±5.1	-19.5±1.6	0.079±0.03
P/Thi-NN-BMMs	891.7±4.9	-25.7±2.5	0.105±0.05

References:

[1] Kanduc M, Kim W K, Roa R, et al. Selective Molecular Transport in Thermoresponsive Polymer Membranes: Role of Nanoscale Hydration and Fluctuations. *Macromolecules*, 2018, 51, 4853–4864.
 [2] Dahal U, Dormidontova E E. Chain Conformation and Hydration of Polyethylene Oxide Grafted to Gold Nanoparticles: Curvature and Chain Length Effect. *Macromolecules*, 2020, 53, 8160–8170.

3. During the introduction English must be improve. It is quite difficult to follow the

discussion. There are some abbreviations that need to be explained during the introduction (i.e. THI) and it has to be extended. No references about what a BBM is. Add some references as it is impossible to consider if they are good candidates for the release of pesticides or not, and to further compare with other porous silica materials. Add references here:

“Double model mesoporous silica materials (BMMs) are a new type of mesoporous material with a worm-like pore of about 3 nm in a double-channel structure and a spherical particle-stacking hole of about 10-30 nm (REF). BMMs are different from single-cell mesoporous materials and have many unique properties, such as controllable structure and controllable particle size (REF).”

What are the main differences between “Double model mesoporous silica materials (BMMs)” and mesoporous silica? As I know mesoporous can be synthesized with a controllable structure and particle size.

Response: We are very grateful for the reviewers' comments. According to the reviewer's kind suggestions, we explained the abbreviations in the introduction of the revised manuscript. The manuscript was polished by native English speakers.

BMMs materials consisted of well-defined small mesopores of 3 nm as well as large interparticles pores. Whereas, the surface area typically lied in the 100-9 m²/g range, and the pore volume varies from 0.25-1.8 cm³/g. Therefore, compared with the traditional mesoporous material, the novel bimodal pore systems of BMMs material allowed the larger drug molecules easier accessibility to the active site due to disappearance of diffused limitation, and thereby improved the desired drug loading and release efficiencies and minimized the blocking influence of mesoporous channels. All these features of BMMs could provide an ideal medium for drug storage and controlled release ^[1,2].

References:

- [1] Gao, L.Sun, J. H.Li, Y. Z. Bimodal Mesoporous Silicas Functionalized with Different Level and Species of the Amino Groups for Adsorption and Controlled Release of Aspirin. *J. Nanosci. Nanotechnol.* 2011, 11: 6690-6697.
- [2] Xiaoqi Jin, Qian Wang, Jihong Sun, Hamida Panezail, Shiyang Bai, Xia Wu, P(NIPAM-co-AA) @BMMs with mesoporous silica core and controlled copolymer shell and its fractal characteristics for dual pH- and temperature-responsive performance of ibuprofen release, *Int. J. Polym. Mater. Polym. Biomater.* 2017, 67: 131-142.

4. I suppose that the release of Thi is performed at different pH as insect have to eat these particles and then the Thi will be released. But, what is a maximum particle size than a small insect can eat? Add this information in the introduction. Further add references here: “The pH of the digestive tract of herbivorous insects is mostly alkaline.”

Response: Most of the pesticides used are currently dispersants and suspending agents with particle size of micron^[1,2]. Compared with conventional pesticides, nano-pesticide has a greater dispersity due to its smaller particle size and better permeability into the epidermis of pests. It has been reported in some literature that the digestive system of insects is alkaline^[3,4], and the pH value was higher than 7 in the sustained-release medium^[5,6]. According to the reviewer's kind suggestions, we added the information and references in the introduction in the revised manuscript.

References

- [1] Chankina OV, Koutzenogii KP, Makarov VI, Kirov EI, Sakharov VM, Zagulyaev GN. 1995 Influence of particle size and concentration, flow rate and precipitation density of aerosol on amount of insecticide deposited on larvae of gypsy moth. *J. Aerosol Sci*, 26, 279-280
- [2] Peter E. Berteau, Wallace A. Deen, Robert L. Dimmick. *Studies of Effects of Particle Size on the Toxicity of Insecticide Aerosols*, 1976, AGRIS/ ResearchGate.
- [3] Khandelwal N, Barbole R S, Banerjee S S, et al. Budding trends in integrated pest management using advanced micro- and nano-materials: Challenges and perspectives. *Journal of Environmental Management*. 2016, 184: 157e169.
- [4] Khandelwal N, Doke DS, Khandare JJ, et al. Bio-physical evaluation and in vivo delivery of plant proteinase inhibitor immobilized on silica nanospheres. *Colloids Surf. B Biointerfaces*. 2015,130: 84e92.
- [5] Li K, Chen K, Wang Q, et al. Synthesis of poly(acrylic acid) coated magnetic nanospheres via a multiple polymerization route. 2019, *Royal Society Open science*. 6, 190141.
- [6] Patra S K, Swain SK. Swelling Study of Superabsorbent PAA-co-PAM/Clay Nanohydrogel. *Journal of Applied Polymer Science*. 2011, 120, 1533-1538.

5. In Figure 2, particles are really aggregated and it is quite difficult to distinguish the spherical morphology. Nanoparticles need to be dispersed and better SEM images (or TEM) need to be performed.

DLS results show that particle size increases from 497 to 891 nm once Thi is loaded and the surface is covered with PAA. Is this increment due to the PAA or Thi in the surface? Comment on that. Is the PAA size related with this difference on size ($891 - 497 = 394$ nm). Do you have the characterization of the BBM+Thi or BBM+PAA combinations?

Response: Thank you so much. Following the suggestion of the reviewer, we conducted the experiments, and better SEM images were obtained (Fig 2).

Fig.2 SEM images of different nanoparticles(A-a ,A-b) BMMs , (B-a ,B-b) NN-BMMs , (C-a ,C-b) P/Thi-NN-BMMs

We supplied experiments and got the characterization of different nanoparticles (Fig. 3). The DLS data (Table 1) showed that the hydrated particle size of BMMs was 497.6 ± 7.8 nm, and the modified particle size of the amino group was further increased to 588.3 ± 4.3 nm. P/NN-BMMs

was used a control, and the particle size was 768.9 ± 5.1 . After loading Thi and surface covering PAA, the hydrated particle size of the nanoparticles increased to 891.7 ± 4.9 nm (Table 1). The synthesis of BMMs with a small pore size of around 3 nm and a large pore size of about 10-30 nm has been performed via post-grafting methods.

Thi was directly loaded into pores during the preparation of P/Thi-NN-BMMs, and a few were adsorbed on the surface of mesoporous silicon. The effect of Thi's loading on the particle size of nanoparticle was not obviously. However, PAA had a major influence on the particle size due to the copolymerization of mesoporous silicon materials on the coating surface.

Fig.3 Particle size distribution of BMMs, NN-BMMs, P/NN-BMMs and P/Thi-NN-BMMs.

6. Important characterization regarding the PAA-BMM coated nanoparticles (without Thi is needed) in order to assess the discussion. Authors need to add TGA, DXRP, DLS of PAA-BBM (without Thi)

Response: Thank you so much. We conducted experiments and added data including TGA, XRPD, DLS of PAA-BBM without Thi. (Fig 4 and Table1)

Fig.4 XRPD pattern(A), TG curves(B), N₂ adsorption/desorption isotherms (C) of samples and corresponding BET surface and pore volume (D).

Table.1 Mean size (nm) , zeta potential(mV) and polydispersity index (PDI) of nano-drug delivery system.

Nps	Mean size (nm)	Zeta potential (mV)	PDI
BMMs	497.6±7.8	15.3±1.5	0.062±0.03
NN-BMMs	588.3±4.3	29.3±2.1	0.098±0.02
P/NN-BMMs	768.9±5.1	-19.5±1.6	0.079±0.03
P/Thi-NN-BMMs	891.7±4.9	-25.7±2.5	0.105±0.05

7. In the TGA indicate where are the 3 weight losses of Thi-NN-BMMS. Adde the TGA of Thi, as it will be easier to compare.

Response: We really appreciate the thoughtful suggestions made by the reviewer. We performed thermogravimetric (TG) analysis and added TGA data in the revised manuscript (Fig4 A).

8. Add DRXP patterns. There is no figure related with this results.

Response: Thank you so much. We carried out the X-ray powder diffraction (XRPD) analysis and added XRPD data in the revised manuscript (Fig 4A).

9. How can the authors calculate the amount of encapsulated Thi? I cannot find this information along the text.

Response: We greatly appreciate the thoughtful suggestions. Based on previous reports, we calculated the loading rate of Thi by TG analysis^[1-3], and added the information in the revised manuscript. Figure 4B showed the results of TG analysis, which provided qualitative and quantitative information about the physical state of P/Thi-NN-BMMs nanoparticles and the control samples. The weight loss of P/Thi-NN-BMMs was more remarkable than those of NN-BMMs and P/NN-BMMs. The significant weight loss of NPs occurred within the temperature

range of 150°C–800°C. The weight loss before 150°C was probably the result of the loss of physically adsorbed water and residual solvent in the channels. The second weight-loss peak occurred at 150°C–290°C, and the weight loss rate of P/ Thi - NN-BMMs NPs was 28% (96.9-68.9%=28%). This weight loss was mainly due to the decomposition of the Thi and terminal amino groups of NN (2.8%) in the mesopores. As a result, the loading rate of P/Thi-NN-BMMs was about 25.2% (28%-2.8%=25.2%)..

References

- [1] Zhang, W B, He S, Liu Y. Preparation and Characterization of Novel Functionalized Prochloraz Microcapsules Using Silica-Alginate-Elements as Controlled Release Carrier Materials. *ACS Appl. Mater. Interfaces*. 2014, 14: 11783-11790.
- [2] Gao Y H, Zhang Y H, He S, et al. Fabrication of a hollow mesoporous silica hybrid to improve the targeting of a pesticide. *Chem. Eng. J*. 2019, 364: 361-369.
- [3] Mainardes R M, Gremião M P D, Evangelista R C. Thermoanalytical study of praziquantel-loaded PLGA nanoparticles. *Rev. Bras. Cienc. Farm*, 2006,42(2):1516-9332.